# Inflammatory Mediators and Pain in Endometriosis: A Systematic Review

**DOI:** 10.3390/biomedicines9010054

**Published:** 2021-01-08

**Authors:** Nikolaos Machairiotis, Sofia Vasilakaki, Nikolaos Thomakos

**Affiliations:** 1Department of Obstetrics and Gynaecology, Accredited Endometriosis Centre, Northwick Park Hospital, London North West University Healthcare, London HA1 3UJ, UK; 2Department of Pharmaceutical Chemistry, School of Pharmacy, University of Athens, Panepistimiopolis Zografou, 157 71 Athens, Greece; svasilak@chem.uoa.gr; 31st Department of Obstetrics and Gynecology, Alexandra Hospital, Gynecologic Oncology Unit, National and Kapodistrian University of Athens, 115 28 Athens, Greece; thomakir@hotmail.com

**Keywords:** inflammation, endometriosis, neuroangiogenesis, pain, cytokines

## Abstract

Background: pain is one of the main symptoms of endometriosis and it has a deleterious effect on a patients’ personal and social life. To date, the clinical management of pain includes prolonged medication use and, in some cases, surgery, both of which are disruptive events for patients. Hence, there is an urgency for the development of a sufficient non-invasive medical treatment. Inflammation is one of the causative factors of pain in endometriosis. It is well established that inflammatory mediators promote angiogenesis and interact with the sensory neurons inducing the pain signal; the threshold of pain varies and it depends on the state and location of the disease. The inhibition of inflammatory mediators’ synthesis might offer a novel and effective treatment of the pain that is caused by inflammation in endometriosis. Objectives: patients with endometriosis experience chronic pelvic pain, which is moderate to severe in terms of intensity. The objective of this systematic review is to highlight the inflammatory mediators that contribute to the induction of pain in endometriosis and present their biological mechanism of action. In addition, the authors aim to identify new targets for the development of novel treatments for chronic pelvic pain in patients with endometriosis. Data Sources: three databases (PubMed, Scopus, and Europe PMC) were searched in order to retrieve articles with the keywords ‘inflammation, pain, and endometriosis’ between the review period of 1 January 2016 to 31 December 2020. This review has been registered with PROSPERO (registry number: CRD42020171018). Eligibility Criteria: only original articles that presented the regulation of inflammatory mediators and related biological molecules in endometriosis and their contribution in the stimulation of pain signal were included. Data Extraction: two authors independently extracted data from articles, using predefined criteria. Results: the database search yielded 1871 articles, which were narrowed down to 56 relevant articles of interest according to the eligibility criteria. Conclusions: inflammatory factors that promote angiogenesis and neuroangiogenesis are promising targets for the treatment of inflammatory pain in endometriosis. Specifically, CXC chemokine family, chemokine fractalkine, and PGE_2_ have an active role in the induction of pain. Additionally, IL-1β appears to be the primary interleukin (IL), which stimulates the majority of the inflammatory factors that contribute to neuroangiogenesis along with IL-6. Finally, the role of Ninj1 and BDNF proteins needs further investigation.

## 1. Introduction

During the menstrual cycle, endometrial-like tissue may spread outside its normal location, the uterus, and be implanted on other locations, more often on pelvic structures to form lesions [1,2]. Consequently, this event activates an immune response, which, in the case of endometriosis, appears to be insufficient for removing these lesions. These lesions attract macrophages, natural killer cells, and cytotoxic T cells [3,4]. The activation of an inflammatory response that follows promotes the secretion of cytokines and chemokines within the peritoneal cavity. This forms a microenvironment that contributes to the development of the ectopic endometrial tissue by facilitating localized angiogenesis and disrupting the normal apoptotic processes [5].

It is a fact that, today, the development of nerve fibers near the endometriotic glands has been documented in all three types of endometriosis (ovarian, deep, and peritoneal) [6,7,8]. The density of the nerve fibers appears to depend on the location of endometriosis, being higher in deep infiltrating endometriosis (DIE) [9,10,11]. It is also known that another characteristic of endometriotic lesions is that they are innervated [8,12,13]. Moreover, endometriotic lesions may contribute to chronic pelvic pain of different intensity, a condition that has a tremendous undesirable effect on patients’ daily life [14].

In general, it is known that an inflammatory response stimulates the secretion of proinflammatory neuromediators [15], promoting neuroagiogenesis [16,17]. This affects the afferent nerve endings of primary sensory neurons, which, in turn, depolarize and transmit the signal by synapsis to the central nervous system (CNS) [10]. This usually results in a low threshold of pain, activating the sensory neurons in a process called ‘peripheral sensitization’ and it produces a nociceptive signal. This signal is transferred to the nerve root of the dorsal horn of the spinal cord and, finally, to the cerebral cortex, via the thalamus and the brain stem [18,19].

However, under chronic inflammatory conditions, such as in endometriosis, alteration on the pain signal will occur due to the persistent inflammatory milieu. The sensitivity to noxious stimuli (hyperalgesia) will change and any future pain will be magnified and form a memory of the unpleasant event in the cerebral cortex. In addition, central sensitization may occur under inflammatory conditions. A couple of studies have proved that central sensitization may be a possible mechanism in the transmission of pain in endometriosis [20,21]. 

Inflammatory pain has been established to be one of the types of pain that occurs in endometriosis [22]. According to the Kyoto protocol of the International Association for the Study of Pain (IASP) basic terminology [23], ‘inflammatory pain’ is the pain that is related to active inflammation and it has been defined as nociceptive pain, although modifications in the nociceptive system may be applied under chronic inflammatory pain. 

In endometriosis, the inflammatory mediators secreted include prostaglandins, vascular endothelial growth factor (VEGF), tumor necrosis factor (TNF-α), nerve growth factor (NGF), and interleukins (IL) [24,25]. Cytokines are signaling proteins, which serve as messengers within the immune system and control a plethora of biological processes, including inflammation and repair, hematopoiesis, innate and acquired immunity, and angiogenesis. They are secreted by many cells and modify the function or phenotype of a cell by binding to its receptors [26].

There are several families of cytokines. The lymphokines’ group was the first identified and characterized as immune-response factors, being synthesized by activated lymphocytes. Other groups include interleukins, which are made by leukocytes to affect other leukocytes (e.g., IL-1β), and chemokines, which demonstrate chemotactic activities and have a special nomenclature (e.g., CXCL12-CXCR7). In addition, there are a number of growth factors that are involved in the proliferation, differentiation, and survival of cells (e.g., VEGF, transforming growth factor-β TGF-β) [26,27]. There are certain types of cytokines, including IL-1β [28], IL-6 [29], and TNF-α [30], which are expressed in locations, such as the dorsal root ganglion [31] and spinal cord [32] or injured nerves [33], and they have been targeted for their involvement in pathological pain. Members of the IL family are known to promote angiogenesis in a synergistic effect [34].

Although the regulation of inflammatory mediators in endometriosis is an active field of research, the mechanism of inflammation-induced pain is unclear, and the profile of cytokines’ expression and their location is contradicting. In this review, we present recent research regarding the expression and role of inflammatory mediators of pain and aim to highlight new targets for the treatment of pain in endometriosis.

## 2. Methods

This systematic review was carried out according to the Preferred Reporting Items for Reviews and Meta-Analyses (PRISMA) Statement.

### 2.1. Data Sources and Search Strategy

Three databases (PubMed, Scopus, and Europe PMC) were searched in order to retrieve articles with the keywords ‘inflammation, pain and endometriosis’, published from 1 January 2016 to 31 December 2020. The search was conducted in 2020. This review has been registered with PROSPERO (registry number: CRD42020171018).

### 2.2. Eligibility Criteria for Articles of Inclusion

A total of 1871 articles were retrieved from the search of the databases. After removing duplicates, 1503 articles remained, which were independently screened by two reviewers (S.V. and N.M.), from which 1373 articles were excluded to identify 130 articles of interest. Of these 130, a further 74 were excluded based on discussion between the authors on the eligibility criteria. The articles were limited to those that were written in English and that presented information on the regulation of inflammatory mediators and related biological molecules in endometriosis and their contribution in the stimulation of pain signal were included. Reviews and letters to the editor were excluded. Any articles describing inflammatory biomarkers of endometriosis, inhibitors of inflammation, hormone receptor antagonists, or the mechanisms involved in immune response were also excluded. 

In order to avoid bias, articles that referred to the regulation of biological molecules in endometriosis were considered, even if their role was not directly linked to the induction of pain, but it was correlated with inflammatory conditions. 

This review is based on 56 remaining articles that fulfilled the eligibility criteria, as shown in Figure 1.

### 2.3. Data Extraction

Data that were extracted from each publication by two of the authors (S.V. and N.M.) included: publication date, authors, population, methods, Exclusion (E)/Inclusion (I) criteria, type of sample, and primary outcome. Table 1 summarizes the recent results in the regulation of inflammatory mediators, as described in this review.

## 3. Results

### 3.1. Biological Role of Key Inflammatory Mediators

In the following paragraphs, we summarize the basic biological functions of the inflammatory mediators that are reviewed further below.

#### 3.1.1. IL-1β Cytokine

IL-1β cytokine is a biomolecule that contributes to cellular signaling via its receptor (IL-1R1). In many chronic inflammatory diseases, an overexpression of IL-1β has been observed. This fact has led to a thorough investigation of its role in inflammation. Its biological role is the encasement of inflammation, of immune responses, and the host defense. IL-1 enhances the production of T lymphocyte-derived cytokines. Consequently, in the absence of IL-1β, the development of tolerance state does not occur, and the immune response is problematic. Moreover, prostaglandin E_2_ and leukotriene B4 function as second messengers in the induction of arachidonate metabolism by IL-1 [52]. The relation of IL-1 to the CNS has also been studied. The expression of the IL-1R1 protein is believed to take place in endothelial cells of the nervous system, under normal and inflammatory conditions. Given the fact that glial cells and neurons can be stimulated by IL-1β, its role in inflammation as well as in neurophysiology remains an open field for debate. 

#### 3.1.2. TNF Cytokine

TNF cytokine is mainly produced by activated macrophages, T lymphocytes, and natural killer (NK) cells. Its biological roles include bond remodeling and the control of infection [53]. In addition, it contributes to leukocyte trafficking and immune complex clearance. TNF-α has been linked with pathological conditions, such as rheumatoid arthritis, atherosclerosis, psoriasis, Crohn’s disease, and sepsis. TNF-α is a key modulator in the synthesis of cytokines and other inflammatory mediators. Interestingly, Yarilina et al. describe the primary macrophages activation after two days of TNF-α stimulation [54]. In this case, TNF-α acts synergistically and it forms an autocrine loop which contributes to the continuous expression of genes encoding inflammatory mediators. This feed-forward loop is able to maintain inflammatory response. In a mouse model, the mRNA of TNF-α, along with COX2, exhibited increased expression in the spinal cord [55]. Although there is a plethora of studies that manifest the role of TNF-α in the promotion of inflammatory diseases, it is also true that anti-TNF-α drugs are not equally effective for all patients and for diseases with a common pathological mechanism. The effect of anti-TNF-α in sepsis has been documented in animal models. It lowers mortality numbers, a positive correlation that has also been observed in patients. However, it has been demonstrated that, in peritonitis, the biological action of TNF-α was necessary for the host defense and the survival of mice.

#### 3.1.3. IL-15 Cytokine

IL-15 cytokine stimulates T-cell proliferation, and this includes the CD4+, γδ, CD8+ and memory subsets. It is also involved in T-cells chemotaxis and interacts with NK cell growth and TNF-α, as well as with other key factors of immune system [56]. Furthermore, IL-15 cytokine has been documented to act as an angiogenic factor in in-vivo models. 

#### 3.1.4. IL-8 Cytokine

IL-8 cytokine, which is also known as neutrophil activating peptide CXCL8, belongs to the CXC chemokine family [57]. It is considered to be a cell migration and chemotaxis factor with a potential role in wound healing. In addition, in pathological conditions, such as cystic fibrosis being related to the pathophysiology of the lungs and other inflammatory diseases, IL-8 demonstrates an upregulation, which makes it an interesting target as a tool for treatment. Interestingly, there are several evidences that IL-8 cytokine facilitates the migration of fibroblasts and endothelial cells [57]. In endometriosis, the IL-8 levels are increased under the synthesis of the oxidative stress products (e.g., 4-hydroxy-2-nonenal) [58]. 

#### 3.1.5. IL-6 Cytokine

IL-6 cytokine is considered to be the promoter of many biological activities. These activities include hematopoietic activity, induction of B-cell, and nerve cell differentiation, while the overproduction of IL-6 may contribute to autoimmune disease, such as rheumatoid arthritis [59]. The ability of IL-6 to manage the survival, proliferation, and also the differentiation of cells is the reason why this interleukin may act as pro- and anti-inflammatory mediator. IL-6 has been found to be involved and control the homeostasis of cell processes, which include the neuroendocrine system, neuropsychological behavior, lipid metabolism, and mitochondrial activities. These findings have made IL-6 a tool to improve the condition of a patient in terms of pain, mood, depression, fatigue, and sleep. Its function also includes the maintenance of the functional integrity of tissues and organs [60]. Under normal conditions, the concentration of IL-6 is relatively low, and it dramatically increases when an event that promotes immune response occurs, e.g., injury or infection. In mouse model, the treatment of an inflamed joint with an antagonist of IL-6 results in the reduction of local inflammation and associated pain [60]. 

#### 3.1.6. IL-32, IL-33 Cytokines

IL-32 cytokine plays an important role in endothelial cell (EC) functions [37]. Increased levels of IL-32 have been detected in a variety of inflammatory diseases. Its pro-inflammatory contribution is the induction of other cytokines and chemokines, including IL-1β, IL-6, and the TNF. Other biological roles of IL-32 include anti-viral and anti-apoptotic properties. The silencing of IL-32 is followed by an incensement of human immunodeficiency virus production.

IL-33 cytokine modulates immune homeostasis and supports tissue repair. It is broadly expressed in stromal and barrier tissue in order to regulate immune responses. Its contribution to pathological conditions, such as diseases of the central nervous system, inflammation, and cancer, has been documented [61].

#### 3.1.7. S1P Cytokines

The biological role of sphingosine 1-phosphate (S1P) is the phosphorylation of sphingosine kinases (SphKs). In the intracellular environment, when the levels of SphK are increased, the level of S1P also follows this trend. S1P affects key mediators in angiogenesis, cancer, and autoimmunity [62]. In addition, S1P facilitates many cell functions when bound to its receptors. Therefore, the action of S1P is determined by the expression of its receptors. 

#### 3.1.8. Chemokines

Chemokines are a group of small size cytokines that induce chemotaxis of monocytes, neutrophils, eosinophils, lymphocytes, and fibroblasts. One of the characteristic functions of chemokines is the induction of leukocytes to the site of inflammation. The concentration of chemokines in the inflamed lesions has been associated with the level of infiltration of leukocyte in the tissue and the level of inflammatory response that they promote [63]. Chemokine fractalkine CX3CL1 belongs to one of the four subgroups of chemokines and may function as both chemoattractant and as an adhesion molecule. Soluble CX3CL1 attracts monocytes and T-cells, and its involvement in angiogenesis and endothelial cell chemotaxis has been documented [64]. Increased levels of CX3CL1 have been detected in the synovial fluid of patients with chronic inflammatory joint disease [63].

#### 3.1.9. Neutrophil Extracellular Traps (NETs)

The neutrophil extracellular traps (NETs) are web-like chromatin structures that are involved in the immune response against infection [65]. NETs prevent the extensive damage of tissues, due to chronic inflammatory conditions and autoimmune disease. Pathological microorganisms and cytokines could be potential stimulators of NETs [65]. 

#### 3.1.10. High-Mobility Group Box (HMGB)

High-mobility group box (HMGB) 1 has a diversity of functions, which include inflammatory cytokines under pathological conditions, signaling regulators in the cytoplasm, and DNA binding (nonhistone) proteins. When it serves as a cytokine, its involvement in immune response and inflammation has been reported as well as chemotaxis and tissue regeneration [66,67]. 

### 3.2. Mediators of Inflammation and Their Expression in Endometriosis 

The products of inflammation (including cytokines and chemokines) are biological molecules with various biological functions making difficult to trace their distinct role to inflammation. Thus, there is a continuous interest for measuring the fluctuation of their regulation in endometriosis, in order to understand the inflammatory process. Although they are part of the inflammatory response, their correlation with the symptom of pain in endometriosis lacks evidence [68]. Up until 2016, research regarding the role of inflammatory mediators in the pathogenesis of disease was limited to only few of them [69]. Therefore, there is continued interest around this complex inflammatory process and the products of inflammation that participate in, with numerous studies being conducted in the past five years (2016 to 2020). The following paragraphs summarize the state-of-the-art in this area of research.

Some genetic causes of inflammation in endometriosis have been found [70]. For example, Gajbhiye et al. concluded that the variance at single nucleotide polymorphisms rs10167914 at the endometriosis risk locus at chromosome 2q13 might influence the genes that are related to the IL-1 family, which are located within 250 kb, and influence the pathogenesis of disease progression [70]. This evidences that inflammatory response in endometriosis is not only driven by the microenvironment of the disease, but there are also genetic causes that may contribute to this. 

Interestingly, the overexpression of proinflammatory cytokines (IL-1β, TNF-α, IL-8), as well as the TNF-stimulated gene-6 factor (TSG-6), in the endometrium of patients with endometriosis was confirmed in a small cohort study conducted by Matteo et al. [35]. The authors reported no alteration in *IL-10* mRNA expression. On the contrary, Gueuvoghlanian-Silva et al. [71] found that *IL-10* mRNA, interferon gamma *IFNG* mRNA, *TGF-β* mRNA, and *IL-7* mRNA expression were importantly increased in deep rectosigmoid lesions. They were also able to identify a correlation between clinical symptoms and genes’ overexpression, which suggested that the expression of genes related to inflammation may depend on the endometriosis location.

In addition, Bellelis et al. [42] reported increased levels of IL-15 concentration in the ectopic endometrium and IL-7 in the eutopic endometrium in patients with DIE. The role of IL-15 in angiogenesis and in the proliferation of human endometrial endothelial cells has been documented before [56,72], offering a possible explanation for the role of IL-15 in endometriosis. 

IL-6 has the potency to contribute to the angiogenesis in cancer and chronic inflammatory diseases according to recently published results [73]. Therefore, its regulation in endometriosis has also been studied. In endometriotic cells (ECs) in endometriosis stage IV, the increased levels of IL-6 showed a dependence on the number of cells, which may explain the high proliferative rate of these cells [40] if we consider the role of this cytokine in cancer. In addition, IL-8 concentration was also significantly higher in the ECs culture, although a moderate correlation was observed with the IL-6 secretion and number of cells [40]. Interestingly, the important role of the endometrial mesenchymal stem cells (eMSC) in the secretion of cytokines was confirmed in reference [40]. It seems that cell communication between ectopic endometrial cells and healthy cells stimulate the alterations in the cytokines expression and may be responsible for the cytokine profile in the peritoneal cavity of patients with endometriosis. Yoshino et al. [41] also suggested that IL-6 protein levels are increased in endometriotic stromal cells (ESC) that are derived from endometrioma due to the function of S1P. IL-1β and TGF-β can both stimulate the expression of *SphK* mRNA, which catalyzes the conversion of sphingosine to the bioactive lipid S1P. These findings show that the synergistic effect between different IL members and the eMSC may also contribute to the regulation of cytokines in endometriosis.

Higher levels of expressed IL-6 were also measured in the peritoneal fluid and serum from women with endometriosis, along with TGF-β isoforms, IL-1β, IL-10, and IL-17AF [36]. These high concentrations of TGF-β in the peritoneal fluid may create suitable conditions for the ectopic lesion formation. In a similar study of reference [39], the expression rate of IL-6 was also higher in patients with endometriosis and it had a dependence on the stage of the disease (higher in III/IV vs. I/II stage). Additionally, steroid hormone fluctuations occurring during the menstrual cycle seem not to affect the IL-6 synthesis. Therefore, it would be interesting to have data on IL-6 levels under estrogens treatment in endometriosis. 

Apart from IL-6, Lee et al. observed a high concentration of IL-32 in the peritoneal fluid in advanced stages of endometriosis [37], while TNF-α and IL-1β concentration were not significantly different, as in case of IL-6. IL-32 had an effect on the viability, proliferation, and invasion of endometrial cells in vitro and increased the size of the ectopic endometrial lesions in mice models. IL-32 possesses angiogenic properties and further studies are needed in order to clarify the mechanism in case of endometriosis, according to recent literature [74].

Regarding IL-33, Miller et al. examined its role in endometriosis [44]. They detected high levels of IL-33 in endometriotic ectopic lesions and enhanced secretion of cytokines (including CXCL1, IL-6, IL-5, IL-7) in plasma as a result of IL-33 stimulation. Locally, IL-33 seems to induce proliferation and growth of the endometriotic lesion. IL-33 has been reported to have both angiogenic and anti-angiogenic function [34]. Due to the synergistic effect, we could conclude that in the case of endometriosis, IL-33 promotes angiogenesis indirectly via the stimulation of other cytokines mentioned above. 

The higher concentration of IL-37 and IL-10 and lower concentrations of IL-17α and IL-2 in the serum of the endometriosis group compared to the control group was reported by Fan et al. [38] without identifying any dependence by the severity of the disease. They also reported a higher concentration of IL-2 in the peritoneal fluid, which may reveal its local effect. In addition, high levels of IL-10 were measured in patients with endometriosis by Nanda et al. [75]. According to their findings, they suggested that its role is to activate matrix metalloproteinase, which further leads to angiogenesis [75]. Regarding IL-17a, Miller et al. [76] also reported elevated levels in the plasma and endometriotic lesions, but they did not connect its role to vascularization.

In contrast, the anti-inflammatory role of IL-37 has been confirmed by Jiang et al. [45] and Kaabachi et al. [46]. IL-37 inactivates the Wnt/β-catenin, serine/threonine kinase p38 mitogen-activated protein kinase (MAPK), extracellular signal-regulated kinase 1/2 MAPK, and JNK/MAPK pathways in vivo experiments. It also inhibits the proliferation, adhesion, migration, and invasion of the endometrial stromal cells and the matrix metalloproteinase (MMP-2 and MMP-9). Its concentration is higher in the peritoneal fluid than in serum in patients with endometriosis. In addition, the expression of *IL-37* mRNA depends on the severity of the disease and demonstrates an inverse correlation with *NF-κB* mRNA expression, which is high in the endothelium tissue of patients [77]. Mancini et al. also confirmed the anti-inflammatory properties for IL-10 [43]. Therefore, the exact role of IL-37 and IL-10 needs more data to be clarified. The regulation of IL-34 has been measured with some interesting finding for its role towards neurogenesis. A link between the inflammatory activation of microglia and neurogenesis has been established in prion disease [78]. The role of colony-stimulating factor 1 receptor (CSF1R) and its ligands (CSF1 and IL-34) in the maintenance of microglial proliferation and survival has also been identified. Given that, the levels of IL-34 are high in the sera of patients with endometriosis and any further stimulation with IL-34 resulted in the up-regulation of CSF1R, p-JAK3, p-STAT6, VEGF, MMP-2, and MMP-9 in eutopic endometrial stromal cells, could lead us to the conclusion that the preconditions for neurogenesis under the disease exist [79]. 

Cytokines of the CXC chemokine family and CXC chemokine receptors have also been studied for their involvement in cell proliferation and invasion in endometriosis. Ruiz et al. [47] studied the expression of CXCR4 and CXCL12 in human endometriosis lesions and in-vitro. They reported that the expression of nuclear CXCR4 depends on the type of lesion, while it enhances the proliferative and migratory potential of the cells. CXCL12 also affects the proliferation, migration, and invasion of endometriotic cells. The CXCR4-CXCL12 and CXCL12-CXCR7 are activated in endometriosis, while CXCR7 is upregulated, depending on the cell type [48]. CXCR4 is expressed by microglia and recruited in the developing cerebral cortex by CXCL12-CXCR4 signaling [80]. Moreover, chemokine fractalkine CX3CL1 and its receptor CX3CR1 are upregulated in ectopic endometrium and they are involved in endometriotic pain conduction, based on the results of the endometriosis animal model [49]. Here, the hypothesis of the central sensitization in endometriosis could be explained by the upregulation of chemokine fractalkine, but this requires further investigation. In addition, CXCL9 level was rather decreased in the serum of patients with endometriosis according to the findings of Ek et al. [50], but further investigation of the underlying mechanism is required. 

Finally, NETs is another inflammatory factor, which has been related to autoimmune and inflammatory conditions. It was detected in the plasma of patients with endometriosis [51]. The increased plasma NETs levels, especially in the DIE group when compared to the control group, suggest the presence of systemic inflammation.

### 3.3. Biomolecules That Stimulate Inflammation and Neuroangiogenesis in Endometriosis

Bioactive molecules, which coordinate with inflammation, have been the subject of several studies, aiming to identify new targets to stop the progression of inflammation. The vasoactive intestinal peptide (VIP) is a pro-angiogenic factor and a systemic immune regulator. Its concentration was increased in serum and peritoneal fluid samples in patients with endometriosis (stage I-III) and symptoms of chronic pelvic pain, when comparing to the control group [81]. This finding demonstrates a link between the VIP pro-angiogenic role and the chronic pelvic pain. Moreover, the modulation of the macrophage phenotype has been proposed in the past to contribute to the endometriosis-associated pain [82]. Their role is still not fully understood, and its function could be pro- or anti- inflammatory, depending on the environment. In this content, Forster et al. [55] documented the direct correlation between the pain score and the expression of insulin-like growth factor-1 (IGF-1) in peritoneal fluid from women with endometriosis. They also observed that linsitinib, which is an IGF-1 receptor inhibitor, had an effect in the pain behavior that was observed in mice with endometriosis. IGF-1 is regulated by endometriosis-associated macrophages and it promotes sprouting neurogenesis and nerve sensitization in vitro. 

The regulation of MAPK by inflammatory mediators in endometriosis has been also studied. The positive correlation between IL-8 and MAPK expression has been observed in endometriotic cells, while the phosphorylation of MAPK is induced by IL-1β and TNF-α [83]. The inhibition of p38 MAPK could suppress the IL-1β and TNF-α-induced IL-8 but it did not affect the endometriotic cell survival. However, the phosho-p38 MAPK levels are affected by the menstrual cycle in endometriosis and this should be taken under careful consideration. In addition, cyclooxygenase (COX-2) dysregulation in ectopic tissues may induce inflammation and MAPKs pathways, in all different stages of the disease, by enhancing the secretion of cytokines in endometriosis [84]. The authors suggested that the inhibition of p38 MAPK may suppress IL-1β-induced COX-2 expression. 

Nematian et al. [85] examined the expression of miRNAs 125b-5p and Let-7b-5p in the serum of patients with endometriosis. They found that the upregulation of miRNA 125b-5p in serum is positively correlated with the upregulation of TNF-α, IL-1β, and IL-6. However, the downregulation of miRNAs Let-7b-5p is negatively correlated to the TNF-α concentration. The authors suggested that these alterations in miRNAs concentration may be a mechanism that contributes to the inflammatory response and further studies require examining whether their inhibition could suppress inflammation. 

The role of HMGB 1 in the activation of innate immunity and inflammation during endometriosis in human samples was also reported [86]. Shimizu et al. found high levels of HMGB 1 in menstrual blood, which could then spread to peritoneal fluid and promote inflammatory conditions via the expression of the receptor for advanced glycation and products (RAGE) in eutopic and ectopic endometrium. In addition, HMGB 1 in complex with lipopolysaccharide induced the expression of *VEGF* mRNA in endometriotic stromal cells. Jaeger-Lansky et al. [87] also found increased levels of HMGB 1 in peritoneal lavage fluid samples from patients. They also observed high levels of IL-6, IL-8, IL-10, and TNF-α levels in these samples, especially in cases of DIE.

Finally, Volpato et al. [88] found that annexin A1, which is a protein that suppresses the phospholipase A2 activity, is lower in the peritoneal fluid of patients with endometriosis. Phospholipase A2 activity is related to arachidonic acid pathway that induces the synthesis of prostaglandins. The authors also measured the expression of some cytokines and found that IL-6 is in a high concentration in the peritoneal fluid, while no significant differences were observed between the patients and the control group for the IL-1β. 

Chang et al. studied two other factors that are correlated with endometriosis [89]. The first is the four jointed box 1 (FJX1), which contributes to angiogenesis. The second is the hypoxia-inducible factor-1a (HIF1a), which contributes to endometriosis progression by enhancing inflammation, angiogenesis, and other processes. The authors found significantly higher levels of the FJX1 protein during the secretory phase, which manifest its dependency by the menstrual cycle in endometriosis. In addition, the positive correlation between FJX1 and hypoxia-inducible factor-1a expression in the eutopic endometrium was documented. FJX1 has also been identified as an inhibitor of dendrite extension [90]; however, such a role in endometriosis still has not been clarified. 

Chen et al. [91] examined the peritoneal fluid of 54 patients with endometriosis to observe the contribution of several inflammatory mediators in the stimulation of myeloid-derived suppressor cells. Consequently, they noted the elevated levels of chemokine (C-X-C motif) ligand 1 and 2 (CXCL1 and CXCL2) already from the disease’s stage I and II, while, in the advanced stage, the concentration of hepatocyte growth factor/c-met (HGF receptor) was remarkably high. The authors concluded that these factors and inflammatory mediators activate myeloid-derived suppressor cells, which contributes to the progress of the disease by altering haematopoiesis and also driving malignancy. This is an additional role of chemokines in the progression of the disease. 

Miyashita et al. [92] introduced an additional link between inflammation and pain, who studied the expression of nerve injury-induced protein 1 (Ninj1) in endometriosis. The authors found that, in endometriotic stromal cells, the *Ninj1* mRNA expression is induced by IL-1β. Nerve fibers occupied the areas of positive staining for Ninj1 in endometriotic lesions. In addition, Ninj1 controls the neural generation. Based on these, the authors suggested that this further illustrates the pathogenesis of pain under these conditions.

In a similar concept, Yu et al. [93] investigated the role of endometrial BDNF in human samples of eutopic endometriosis stromal cells in DIE lesions. They found that IL-1β stimulates the synthesis of BDNF isoforms, and several biological pathways contribute to this (NF-*κ*B, JNK and mTOR). These findings support the hypothesis that proinflammatory factors in the microenvironment of endometriotic cells promote neuroangiogenic milieu via neuroinflammation. In addition, high levels of plasma BDNF were measured in women with ovarian endometrioma, but not for DIE or peritoneal endometriosis. The BDNF levels showed a dependence on the severity of pelvic pain [94]. 

## 4. Discussion

The pathophysiology of endometriosis involves biological mechanisms that induce pain. Although there are numerous studies on the related biological processes, there has not been any success in developing an effective treatment for this devastating symptom, which can have a dramatic effect on a patient’s life. The first-line medical treatment includes hormonal agents that suppress endometriosis, leading to the alleviation of pain. As an alternative therapy, the regulation of inflammatory mediators poses as a promising non-hormonal-based option.

Inflammatory mediators in endometriosis have an active role in the induction of pain via angiogenesis and neuroangiogenesis. There are several biological pathways for this and, although the biological mechanisms are not fully understood, new data are continually revealed. Angiogenesis, which refers to the growth of blood vessels, is an essential response to wound healing under inflammation. Especially in endometriosis, fibrogenesis has been accused as the leading force for the malignant transformation of the disease [95]. The growth of nerve fibers (neuroangiogenesis) in parallel to angiogenesis is the significant event that has been observed in endometriosis, which provides a direct link between peritoneal endometriotic lesions and sensory nerve fibers [7]. The relation between angiogenesis and neuroangiogenesis is well established in a cerebrovascular accident [96]. Further studies on this have been accomplished, showing the influence of inflammation on alteration of the pain signal. Peripheral endometriosis-like lesions appear to cause adaptations in central glial reactivity, according to Dodds et al. [97]. This may lead to central sensitization, which helps in the establishment of inflammatory conditions via neurogenic inflammation. 

Therefore, targeting inflammatory mediators could result in an efficient treatment of pelvic pain in endometriosis. Hitherto, TNF-α, PGE_2_, and MIF (human macrophage migration inhibitory factor) have been proposed for the suppression of inflammation in endometriosis [69]. In this review, we focused on the latest research between 2016 and 2020 on the role of inflammatory mediators and biological molecules forming an inflammatory response, in order to identify the key targets and, thus, alternate options for the treatment of inflammation-induced pain in endometriosis. 

It is a known fact that inflammatory mediators cause peripheral sensitization and trigger the CNS in multiple ways. These include the activation of afferent nerves [98], the cytokine transporters at the blood-brain barrier [99], and the activation of IL-1 receptors on perivascular macrophages and endothelial cells of brain venules, which locally promote prostaglandin E_2_ secretion [100]. Hence, the profile of cytokines in patients with endometriosis has been studied in order to understand the mechanism whereby inflammatory conditions promote pain in endometriosis. 

Based on the reviewed articles, we would like to highlight some biological molecules that create a link between the induction and the severity of pain in endometriosis. The proinflammatory interleukin, IL-6, has been identified in high concentrations in endometriosis [101]; however, some of the contradicting studies exist. In this review, four studies showed that IL-6 is overexpressed in endometriosis and the level of its expression depends on the disease’s severity. In addition, the expression of IL-1β, which has been associated with the risk of endometriosis [102] appears to depend on the stage of the disease and stimulates the upregulation of biomolecules that induce pain, as mentioned above. 

Moreover, the CXC chemokine family and CXC chemokine receptors, along with chemokine fractalkine CX3CL1 and its receptor CX3CR1, have also been highlighted for their contribution in the development of the local inflammatory milieu and the formation of the peripheral hyperalgesia in endometriosis. Overall, CXC chemokines have been identified as the link between inflammation and angiogenesis and, especially, CXCL9 has been shown to promote liver fibrosis in animal models [103,104]. Thus, the upregulation of these molecules in endometriosis is in accordance with the fibrogenesis event. Therefore, their inhibition should be explored as a possible way to suppress fibrogenesis, which could be the primary level of neuroangiogenesis and the induction of pain. 

An alternative mechanism of inflammation-induced pain in endometriosis was introduced by Berkley et al. [105,106], who observed the development of an autonomic and sensory innervation that formed by ectopic cysts. Subsequently, the inflammation-induced neuroangiogenesis in endometriosis has been studied and recently demonstrated in a mouse model [107]. In accordance with these findings, here we present two studies that identified the upregulation of factors that assist neuroangiogenesis. 

Firstly, the study by Yu et al. [93] verified that the upregulation of BDNF precursors, dimers, and mature proteins depends on IL-1β action through kinases pathways (JNK, NF-*κ*B, and mTOR signaling). BDNF, along with NGF and neurotrophins (NT-3, NT-4/5), have been identified in samples of ovarian endometriomas in advanced endometriosis. Additionally, the authors showed the changes that were induced by IL-1β on the morphology of eutopic endometriosis stromal cell (EESC) and its effect on RANTES secretion via the IL-1 receptor. BDNF has pivotal roles in the maintenance of neurons in the CNS and it is involved in neuroangiogenesis. Secondly, Miyashita et al. [92] proved the regulatory role of IL-1β in the expression of *Ninj1* mRNA in endometriotic stromal cells and the presence of nerve fibers in lesions with positive staining for Ninj1. Ninj1 is a novel neurotransmitter and, along with other activities, it appears to enhance the innervation under the known inflammatory conditions that are present in endometriosis lesions [108,109]. 

Finally, the expression of prostaglandins E_2_ (PGE_2_ and PGF_2_α) is an additional vector that regulates the inflammation-induced pain in endometriosis. McAllister et al. [110] found an association between the two events: the severity of endometriosis induced vaginal hyperalgesia and the regulation of PGE_2_ and PGF_2_α in the peritoneal cavity in rodents. When a non-selective cyclooxygenase inhibitor was used in order to treat rats, a reduction of vaginal hyperalgesia was observed. Moreover, measurements of the cyst sensory innervation density showed a level of dependence by PGE_2_. It is known that the concentration of PGE_2_ in the peritoneal fluid is higher for women with endometriosis and it contributes to the survival and growth of endometriosis lesions [111,112]. In addition, it affects the peripheral nociceptors, increasing the response to peripheral stimuli via transient receptor potential cation channel subfamily V member 1 and stimulating the chronic inflammatory pain by interacting with prostaglandin E_2_ receptors EP2/EP4 [113,114,115]. Therefore, the inhibition of PGE_2_ receptors has been studied for the treatment of inflammation-induced pain in endometriosis [116,117], with some interesting results in diminishing angiogenesis and the innervation of endometriotic lesions. As shown in a rat model, EP1/EP4 receptors are involved in PGE_2_-induced BDNF synthesis in injured dorsal root ganglion neurons, manifesting the role of PGE_2_ in the development of neuropathic pain [118,119,120]. These promising results should direct the current research towards the development of selective inhibitors that could suppress the synthesis of PGE_2_ in endometriosis in order to better understand its effect in the induction of chronic pelvic pain. 

The inhibition of inflammatory mediators and relative biological molecules have been studied, aiming to suppress angiogenensis and neuroangiogenesis in endometriosis and reduce the symptom of pain. Although, a number of drugs, including sunitinib, thalidomide, and rosiglitazone, have been developed targeting VEGF, TNF-α, NGF, and interleukins [121]. The disadvantage of these drugs is that they may inhibit the physiological angiogenesis, which is crucial for reproductive function and wound healing, and it increases the risk of teratogenicity. Contrarily, even though new anti-inflammatory molecules have been developed [122,123,124], there is a lack of information regarding their effect on the symptom of pain. This type of information is important in understanding the role of inflammatory mediators in endometriosis and pain.

Although there are readily available treatments, targeting cytokines, for pain symptoms, there is a major factor that affects their effectiveness. We need to focus on defining the dominant cytokine in the disease process in order to use the inhibition of cytokines as a tool to suppress an inflammatory response and relative pain in endometriosis. Based on clinical studies, we know that, although cytokine targeted therapies have been designed for diseases that share common mechanisms of pathophysiology, the efficacy of the therapy depends on the particular conditions. For example, tocilizumab, which blocks the IL-6 receptor, is an effective treatment for arthritis, but not for other choric inflammatory conditions, such as psoriasis. Therefore, it is not enough to measure the regulation of cytokines in endometriosis. Further experiments need to identify the dominant cytokines that orchestrate the inflammatory response under endometriosis. IL-1 could play a role under this condition (Figure 2), but further experiments are required in order to prove this.

## 5. Conclusions

Inflammatory mediators are molecules with multiple biological functions, and their role in the induction and amplification of pain in endometriosis is complicated. The inhibition of inflammatory mediators as an approach for treating pain is a promising alternative to hormonal-depending medication that is widely used today. Novel factors that are stimulated by inflammatory mediators and contribute to neuroangiogenesis have been recently identified, including Ninj1 and BDNF. The role of FJX1 in endometriosis needs further investigation. IL-1β seems to be the primary interleukin that stimulates most of the factors that contribute to neuroangiogenesis, along with IL-6, the CXC chemokine family, and chemokine fractalkine. The secretion of inflammatory mediators works as a response to the stimulation by macrophages and biological molecules, including estrogen and the mediators themselves. This process is a loop of events that is not easy to break. Therefore, a multi-target therapy seems to be the most promising approach in the treatment of pain in endometriosis and research should focus on this direction. 

## Figures and Tables

**Figure 1 biomedicines-09-00054-f001:**
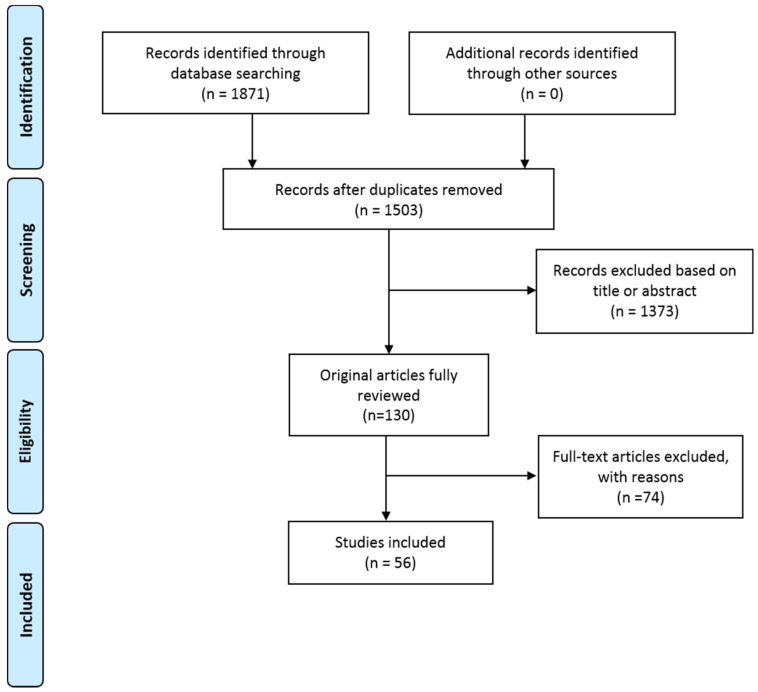
PRISMA flow diagram.

**Figure 2 biomedicines-09-00054-f002:**
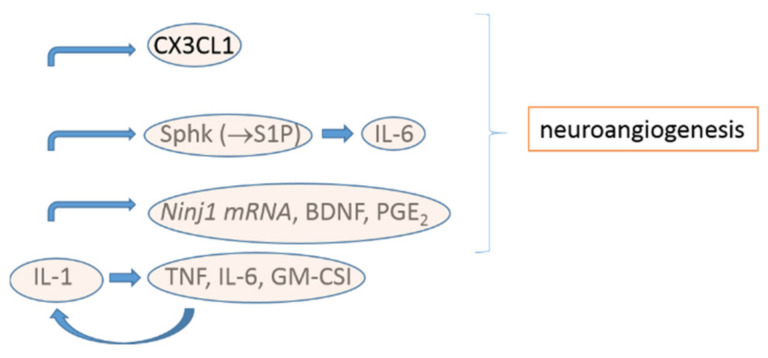
Proposed mechanism of stimulation of inflammatory mediators by IL-1 in endometriosis, as presented in this review.

**Table 1 biomedicines-09-00054-t001:** Regulation of inflammatory mediators that are described in this review.

Cytokines-Chemokines	Expression in Endometriosis
IL-1β	Overexpression in endometrium and the peritoneal fluid [35,36]-no significantly different in the peritoneal fluid in advance stage [37]
IL-2	Lower expression in serum [38]
IL-6	Higher expression the peritoneal fluid and serum-dependence on the severity [36,39,40,41]
IL-7	Higher expression in eutopic DIE [42]
IL-8	Overexpression in the endometrium [35,40]
IL-10	Higher in the peritoneal fluid and serum/anti-inflammatory properties [36,38,43]
IL-15	Higher in ectopic DIE [42]
IL-17A	Lower in serum [38]/IL-17AF higher in the peritoneal fluid and serum [36]
IL-32	High concentration in the peritoneal fluid-depend on the severity [37]
IL-33	Higher levels in ectopic lesions [44]
IL-37	Higher expression in serum—anti-inflammatory role [38,43,45,46]
TNF-α	Overexpression in endometrium [35]
CXCR4 (CXCR4-CXCL12 activation)	Depends on the type of lesion [47]
CXCR7 (CXCL12-CXCR7 activation)	Upregulation in ectopic glands-depending on cells type (epithelial > stromal) [48]
CX3CL1/CX3CR1	Upregulation CX3CL1 on macrophages and CX3CR1 on myelin sheath of sciatic nerve fibers [49]
CXCL9	Decrease levels in serum [50]
NET	Increased in plasma [51]
TGF-β	Increased in the peritoneal fluid and serum [36]

## Data Availability

Data sharing not applicable.

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
