# Peer review of "Inflammatory Mediators and Pain in Endometriosis: A Systematic Review"

_biomedicines, 2021, doi:10.3390/biomedicines9010054_

Round 1

Reviewer 1 Report

The aim of this systematic review is to review the inflammatory mediators which contribute to the induction of pain in endometriosis and present their biological mechanisms. in order to develop new medical therapies. The review is well conducted, following Prisma guidelines, and references updated, but a major revision is needed before accepting the manuscript for publication in Biomedicines.

Comments.

  • In the “Results” the authors conduct an accurate analysis of mediators’ biological roles generally involved in inflammation. However, the description of the mediators' action should be less verbose and perhaps inserted in a separate paragraph in the context of the introduction. I would suggest adding a table to facilitate the reading comprehension.
  • Although the analysis of cytokines involved in endometriosis is very thorough, it is somewhat dispersive. In the relative paragraph, the review’s results on mediators may be described by dividing them on the basis of the cytokines found in each study, and their location (i.e., peritoneal fluid, serum, DIE, endometriomas etc.)
  • It could be useful to review and include some data on the role of the mediators involved in the oxidative system which is often compromised in the disease (see the article by Scutiero G, at al. Oxidative Stress and Endometriosis: A Systematic Review of the Literature. Oxid Med Cell Longev. 2017;2017:7265238.), and on the available molecules with antioxidant activity such as curcumin or N-Acetyl-L-cysteine (Pittaluga E, et al. More than antioxidant: N-acetyl-L-cysteine in a murine model of endometriosis. Fertil Steril. 2010 Dec;94(7):2905-8; Porpora MG et al. A promise in the treatment of endometriosis: an observational cohort study on ovarian endometrioma reduction by N-acetylcysteine. Evid Based Complement Alternat Med. 2013;2013:240702.)
  • In the discussion, the presence of new targets for therapeutic approaches could be deepened; the authors should specify the new anti-inflammatory molecules available for endometriosis treatment, rather than just mentioning them (line 30). Moreover, in literature there are some studies on the use of immunomodulators, that should be cited (i.e. Kotlyar Aet al. Use of immunomodulators to treat endometriosis. Best Pract Res Clin Obstet Gynaecol. 2019 Oct;60:56-65).
  • It is required a minor revision of English language

Author Response

Thank you very much for your comments

1)We briefly mention the mediators. They have been extensively reviewed elsewhere.

2) We appreciate your expertise and we have considered them carefully.

3) The aim of this review is to report the latest of the regulation of
inflammatory mediators and related biological molecules in endometriosis
and their contribution in the stimulation of pain signal. N-acetylocysteine
and curcumin are not approved treatments for endometriosis and the
inflammatory-induced pain.

4) In the literature, it is not clear if new anti-inflammatory molecules,
targeting the inflammatory mediators mentioned in the review, are
available for endometriosis treatment. In the manuscript we mention some
known drugs against angiogenesis, not specific for endometriosis.
The aim of this review is to summarize the latest biological findings of
inflammatory mediators in endometriosis, in a hope to identify new targets
for therapeutic approaches.
As for immunomodulators, the immune response is not within the scope of
this review ( 2.3. Eligibility Criteria '.... or mechanisms involved in
immune response were also excluded'  )

Reviewer 2 Report

This manuscript intended to examine inflammatory mediators and pain in endometriosis. The amount of concerns regarding grammar, scientific language not used, and the limit of references from 2017 to 2019 prevented me from fully reviewing this manuscript. The idea for the article is wonderful, but in the current state (with errors, font sizes different, references not enclosed in sentences, some sentences not referenced), this is just a draft and must be fully reworked before reviewers see the manuscript again. 

Major concerns: 

more data than 2017 to 2019 should be included for a review article unless this is an "update" of a previously published review. 

Angiogenic inhibitors have been tried and do not work well in animal models of disease. Additionally, these agents would be quite toxic to women and shut down many vital processes more so than endometriosis. 

Regarding pain sensitivity- women with endometriosis have quite high pain thresholds. Adding an additional stimulus to an already high pain threshold may seem that these women have a diminished response; however, if you are already in an increased state, the level to reach max is much closer than a person who has no pain at all. The relatively of this is critical and must be pointed out as it is not typically done. 

Author Response

The systematic review has been fully reworked and edited in the journal's template

1) We decided to include only the period between 2017-2019 because there were reviews before 20127 and we would like to focus only on recent studies

2) line 465. We also report that there are disadvantages of such inhibitors. However, whether it is more or less than endometriosis itself we can not say yet since there is no specific treatment with such inhibitors and corresponding clinical data. 

3)Apart from the surgical treatment and given the risk of recurrence it is fundamental to search for new factors  in order to control the pain. The inhibition of inflammatory mediators’ synthesis might offer a novel and effective treatment for alleviating the pain caused by inflammation in endometriosis.

Round 2

Reviewer 1 Report

The Authors address most of the reviewer's comments.

The manuscript can be accepted for publication 

Author Response

We would like to thank you for your kind indications for improving this manuscript . We have applied major revisions in the text. We have tried to reorganize and improve the way things are presented in the text and make the meaning of the context more comprehensive. English grammar and spelling has been checked by native speakers and corrected accordingly.

Reviewer 2 Report

The enclosed manuscript was significantly updated from the first version; however, there are still quite a few issues with this manuscript. 

  1. Justification for being just 2 years. Now, it is the end of 2020 and the review is 2017-2019. Concern that this narrow window is missing a lot of previous work and the figure with mechanism is focused on the 2 years of work and not fully encompassing previous work. 
  2. Scientific and English language are needed in many places. The use of alpha for instance with TNFa is written a or alpha in the same paragraph. This just makes the reviewer upset that you didn't proof read the manuscript before sending it for review. 
  3. The article is built on many review articles. In many cases, it is a review article of review articles. The primary literature is not cited. Furthermore, there are a lot of places that need to include references with the statements being made. 
  4. The article can be broken down into more sub-categories. For instance, have a heading under the inflammatory mediators, ie. IL6, IL61beta, etc. In many cases, the manuscript is disjointed and jumps from one topic to another and doesn't flow well. For instance, Line 507 to 528. The Moreover paragraph does not flow. Additionally, paragraphs are short and can include more in the discussion than short choppy paragraphs. 

Author Response

We would like to thank you for your kind indications for improving this manuscript. We have applied major revisions in the text in order to address your comments. As such:

comment 1: Justification for being just 2 years. Now, it is the end of 2020 and the review is 2017-2019. Concern that this narrow window is missing a lot of previous work and the figure with mechanism is focused on the 2 years of work and not fully encompassing previous work. 

answer:

This review describes the latest experimental results of the biological mechanisms which possible correlate with pain in endometriosis and involve the activity of cytokines. Original articles which included in the present review studied cytokines regulation in tissue samples of patients with endometriosis, not in animal models.

Within the period of three years 2017-2019 relative work on this specific topic has been published to a great extent comparing to earlier years. Up until 2016 there were data only for a few mediators of inflammation in endometriosis according to the review article ‘Nothnick  et al. Recent advances in the understanding of endometriosis: the role of inflammatory mediators in disease pathogenesis and treatment. F1000Research  2016.

We have now expand the reviewed period 2016-2020.

We added the following references :

[75] Cytokines, Angiogenesis, and Extracellular Matrix Degradation are Augmented by Oxidative Stress in Endometriosis

[76] IL-17A Modulates Peritoneal Macrophage Recruitment and M2 Polarization in Endometriosis

[58] Amine oxidase 3 is a novel pro-inflammatory marker of oxidative stress in peritoneal endometriosis lesions

 [76] Involvement of angiotensin II receptor type 1/NF-kB signaling in the development of endometriosis

More specific, according to Scopus database (24/12/2020) the keywords ‘inflammation  AND pain  AND endometriosis’ returned:  40 documents published in 2015, 45 documents published in 2014, 32 documents published in 2013, 32 documents published in 2012 and none of the original articles could be directly correlated with the concept of the present review. According to Scopus database (24/12/2020), the keywords ‘inflammation  AND pain  AND endometriosis’ returned 65 articles published in 2020. None of them could be directly correlated with the concept of the present review. According to PubMed: 538 articles published in 2020, we refered 4 papers in the review which were within the scope. 292 articles published in 2016 we had already reviewed 4 papers.

Figure 2 presents results that are included in this review. To avoid misunderstandings we added in the title: ‘Figure 2. Proposed mechanism of stimulation of inflammatory mediators by IL-1 in endometriosis, as presented in this review.’

comment 2: Scientific and English language are needed in many places. The use of alpha for instance with TNFa is written a or alpha in the same paragraph. This just makes the reviewer upset that you didn't proof read the manuscript before sending it for review. 

answer: English grammar and spelling has been checked by native speakers and corrected accordingly.

line 169: … It is considered to be a cell migration …

line 221: ..group box (HMGB) 1  have has…

line 395: Two other factors which are correlated …

Typo mistakes were corrected, including TNF-α at lines 148,149,150,153,238,358,359,502

comment 3: The article is built on many review articles. In many cases, it is a review article of review articles. The primary literature is not cited. Furthermore, there are a lot of places that need to include references with the statements being made. 

answer: Review articles have been mainly used in paragraph ‘3.1. Biological role of key inflammatory mediators’ to explain briefly (as a short introduction) the biological role of the biological molecules mentioned in the followed two paragraphs. The reviewed original articles are in paragraphs ‘3.2. Mediators of inflammation’ and ‘3.3. Biomolecules that stimulate inflammation and neuroangiogenesis in endometriosis’.

In order to make this more clear in the text, we added:

line 131:  ‘In the following paragraphs, we summarize the basic biological functions of the inflammatory mediators which are reviewed further below.

line 234: Although they are part of the inflammatory response, their correlation with the symptom of pain in endometriosis lacks evidence [68]. Up until 2016, research around the role of inflammatory mediators in the pathogenesis of disease was limited to only few of them.[69] Therefore, there is continued interest around this complex inflammatory process and the products of inflammation which participate in, with numerous studies being conducted in the past five years (2016 to 2020). The following paragraphs summarize the state-of-the-art in this area of research.

References are usually placed in the beginning or in the end of each paragraph in the text. To avoid misunderstandings, we added- repeated the references within the paragraph throughout the text.

comment 4: The article can be broken down into more sub-categories. For instance, have a heading under the inflammatory mediators, ie. IL6, IL61beta, etc. In many cases, the manuscript is disjointed and jumps from one topic to another and doesn't flow well. For instance, Line 507 to 528. The Moreover paragraph does not flow. Additionally, paragraphs are short and can include more in the discussion than short choppy paragraphs. 

answer: There are not a lot of studies yet that investigate the cytokines role in the induction of pain in endometriosis. Therefore, the data available are not easy to get together. The way it is, usually is one or two studies on each biomolecule, which makes it difficult to group them together.

However, we have applied major revisions in the text. We have tried to reorganize and improve the way things are presented in the text in order to make it flow better and avoid gaps in the meaning of the context, by connecting paragraphs and adding sub-heading and extra phrases in order to clarify the meaning of the text.

We added sub-headings:  3.1.1-3.1.10

We added :

line 234: “The following paragraphs summarize the state-of-the-art in this area of research.” 

line 242: “This evidences that inflammatory response in endometriosis is not only driven by the microenvironment of the disease but there are genetic causes that may contribute to this.”

line 256: “offering a possible explanation for the role of IL-15 in endometriosis’

line 261: ‘if we consider the role of this cytokine in cancer’

line 279: ‘Therefore, it would be interesting to have data on IL-6 levels under estrogens treatment in endometriosis’

line 307: ‘Therefore, their exact role of IL-37 and IL-10 needs more data to be clarified.’

line 309: ‘Regulation of IL-34 has been measured with some interesting finding for its role towards neurogenesis.’

line 327-334: reorganize the paragraphs and added ‘In addition, CXCL9 level was rather decreased in the serum of patients with endometriosis according to the findings of Ek et al., but further investigation of the mechanism underline required.’

line 344-353: connected paragraphs with similar context  

line 335: ‘Finally, another inflammatory factor is  NETs which has been related to autoimmune and inflammatory conditions. It was detected in the plasma of patients with endometriosis.’

line 365: ‘The regulation of MAPK by inflammatory mediators in endometriosis has been also studied’

line 363: reorganize the paragraph to : ‘The regulation of MAPK by inflammatory mediators in endometriosis has been also studied. The positive correlation between IL-8 and MAPK expression has been observed in endometriotic cells, while phosphorylation of MAPK is induced by IL-1β and TNF-α [80]. Inhibition of p38 MAPK could suppress the IL-1β and TNF-α-induced IL-8 but it did not affect the endometriotic cell survival. However, phosho-p38 MAPK levels are affected by the menstrual cycle in endometriosis and this should be taken under careful consideration. In addition, cyclooxygenase (COX-2) dysregulation in ectopic tissues may induce inflammation and MAPKs pathways, in all different stages of the disease, by enhancing the secretion of cytokines in endometriosis [81]. Authors suggested that inhibition of p38 MAPK may suppressed IL-1β-induced COX-2 expression.

line 390: ‘Phospholipase A2 activity is related to arachidonic acid pathway which induces the synthesis of prostaglandins.’

line 472: Therefore, their inhibition should be explored as a possible way to suppress fibrogenesis, which could be the primary level of neuroangiogenesis and the induction of pain.   

line 507: ‘These promising results should direct the current research towards the development of selective inhibitors that could suppress the synthesis of PGE2 in endometriosis to better understand its effect in the induction of chronic pelvic pain.’

 line 544: Therefore a multi-target therapy seems to be the most promising approach in the treatment of pain in endometriosis and research should focus toward this direction.

‘Line 507 to 528 The Moreover paragraph..’  The numbers are different in the manuscript. However, the paragraph in the Discussion section that starts with ‘Moreover, the CXC chemokine family and CXC chemokine receptors …’ refers to the role of chemokine family, which are small cytokines, in the induction of pain. In the discussion we have summarized the role of inflammatory mediators (interleukins, cytokines, prostaglandins) in the induction of pain and so the paragraph is in the context of text. In order to clarify this and make it more comprehensive we added:

line 516-518: ‘Thus, upregulation of these molecules in endometriosis is in accordance with the fibrogenesis event. Therefore their inhibition should be explored as a possible way to suppress fibrogenesis, which could be the primary level of neuroangiogenesis and the induction of pain.’